# Non-Invasive MRI of Blood–Cerebrospinal Fluid Barrier Function

P. G. Evans[1], M. Sokolska [2], A. Alves[3], I. F. Harrison[1], Y. Ohene[1], P. Nahavandi[1], O. Ismail [1], E. Miranda[3], M. F. Lythgoe[1], D. L. Thomas [4,5] & J. A. Wells [1✉]

The blood–cerebrospinal fluid barrier (BCSFB) is a highly dynamic transport interface that serves brain homeostasis. To date, however, understanding of its role in brain development and pathology has been hindered by the absence of a non-invasive technique for functional assessment. Here we describe a method for non-invasive measurement of BSCFB function by using tracer-free MRI to quantify rates of water delivery from arterial blood to ventricular cerebrospinal fluid. Using this method, we record a 36% decrease in BCSFB function in aged mice, compared to a 13% decrease in parenchymal blood flow, itself a leading candidate biomarker of early neurodegenerative processes. We then apply the method to explore the relationship between BCSFB function and ventricular morphology. Finally, we provide proof of application to the human brain. Our findings position the BCSFB as a promising new diagnostic and therapeutic target, the function of which can now be safely quantified using non-invasive MRI.

[1] UCL Centre for Advanced Biomedical Imaging, Division of Medicine, University College London, London, UK. [2] Medical Physics and Biomedical Engineering, University College London Hospitals NHS Foundation Trust, London, UK. [3] Pathology Core Facility, University College London Cancer Institute, London, UK. [4] Neuroradiological Academic Unit, Department of Brain Repair and Rehabilitation, UCL Queen Square Institute of Neurology, London, UK. [5] Leonard Wolfson Experimental Neurology Centre, UCL Queen Square Institute of Neurology, London, UK. ✉email: jack.wells@ucl.ac.uk

The brain relies on continuous perfusion by the peripheral circulation to sustain normal function. There are two distinct transport interfaces separating blood from brain: the blood–brain barrier (BBB—the blood vessels within the parenchyma) and the blood–cerebrospinal fluid barrier (BCSFB—the choroid plexuses (CP) in the ventricles). The structure of the BCSFB differs markedly to the BBB and thus represents a unique arbitrator of molecular and cellular transfer between blood and brain[1]. Importantly, the BCSFB drives the bulk flow of cerebrospinal fluid (CSF) via the net secretion of water across the CP[2]. These actions facilitate the delivery of numerous compounds that have diverse roles in maintaining normal brain function (e.g., $Ca^{2+}$, insulin-like growth factor 1[3,4]) as well as the clearance of substances that may be detrimental if allowed to accumulate in the CNS, such as amyloid beta[1,5]. This functionality is supported by the vast surface area of the BCSFB, approximately one-half that of the BBB[6].

Given these unique capabilities, BCSFB dysfunction is now suspected to underlie a wide pathophysiological spectrum[7,8]. For example, impairment to CSF-mediated brain clearance may initiate neurodegenerative cascades defined by the accumulation of toxic proteins, as observed in Alzheimer's disease[9–11]. Moreover, as a key site of immune cell entry into the CNS, the BCSFB may be central to the development of autoimmune conditions such as multiple sclerosis[12,13]. Currently, however, our understanding of BCSFB function in disease and therapy is restricted by the absence of a non-invasive measurement technique. Indeed, the number of studies centred on measurement of BCSFB function is miniscule compared with measures that probe BBB function using non-invasive imaging techniques such as blood oxygen level dependent (BOLD) functional MRI and arterial spin labelling (ASL) MRI.

Here, we open a new window into the study of brain physiology by introducing a non-invasive, tracer free, technique for the quantitative assessment of BCSFB function. We describe a translational MRI method that measures the rate of delivery of arterial blood water across the BCSFB, into ventricular CSF. We begin with characterisation and validation: (1) we demonstrate that the method produces images of blood-to-CSF water delivery co-localised to the site of the CP; (2) we capture the dynamic time-course of blood water delivery across the BCSFB and use an adapted kinetic model, together with histologically derived estimates of CP mass, for quantification of CP blood flow; (3) using controlled modulation of ASL labelling efficiency and a hybrid diffusion-ASL MRI sequence, respectively, we confirm that the source of the novel BCSFB functional signal derives from endogenous arterial blood water that has been delivered to ventricular CSF; (4) we demonstrate the sensitivity of the technique to detect specific downregulation of BCSFB function with administration of the anti-diuretic hormone vasopressin.

Given histological and in vitro data linking BCSFB dysfunction to age-related cognitive decline[14,15], the technique introduced here represents a promising candidate to meet the urgent clinical need for early predictive biomarkers of future neurodegenerative outcome. Therefore, next, we applied the method to aged mice, with comparison to adult control mice. Measures of cortical blood flow were also recorded, a sensitive marker of age-related neurodegenerative processes[16].

The morphology of the lateral ventricles remains a defining feature of hydrocephalus and is increasingly recognised as a radiological marker of neuropsychiatric disorders such as autism[17]. Ventricle size is thought to be determined by a balance between CSF secretion and absorption[1]. The precise mechanisms, however, that regulate ventricle size in the developing and diseased brain remain unknown, largely because there are no techniques able to capture meaningful correlates of CSF secretion or

absorption that do not require brain surgery. Given that the BCSFB is the main locus of CSF secretion[1], here we examine the possible utility of the new MRI method to capture the action of the BCSFB to shape ventricular morphology. Supported by scan–rescan reproducibility data, we investigate correlations between our measure of BCSFB function and ventricular volume, in turn, investigating whether the quantitative estimates of BCSFB function reported here represent a meaningful correlate of CSF secretion. Finally, we conclude by providing proof of application of this translational technique to the human brain at 3T.

This quantitative method represents a novel tool to progress understanding of BCSFB function in brain development and pathology which, in turn, may lead to new diagnostic and therapeutic strategies that target the BCSFB in conditions such as Alzheimer's disease and hydrocephalus.

## Results

**Non-Invasive MRI of BCSFB Function: Characterisation and Validation.** The method is based on the established principles of ASL MRI but with an ultra-long echo time acquisition (220 ms @ 9.4 T) that ensures only signal from CSF compartments, which have a long T2 value, are captured within the ASL images (Supplementary Fig. 1). Thus, in this non-invasive technique, blood water in the brain's feeding arteries is magnetically labelled and after a given inflow time (TI), an image is acquired of labelled blood water that has travelled via the BCSFB and into ventricular CSF (Fig. 1a). Acquiring this novel, putative, signal of BCSFB function (termed the BCSFB-ASL signal) in the lateral ventricles at multiple TIs revealed a dynamic time-course markedly different to the traditional ASL signal that probes BBB function (BBB-ASL signal), reflecting, in part, the increased arrival time and decreased rate of T1 decay of the labelled water molecules that are delivered to the CSF (Fig. 1b). Given the known location of the choroid plexus at the caudal aspect of the lateral ventricles[18] (confirmed by histological assessment in the same mice), we compared the BCSFB-ASL signal at the rostral and caudal sections of the lateral ventricles (Fig. 1c), recording a significant reduction in the rostral region as hypothesised ($p < 0.0001$). This demonstrates that the BCSFB-ASL signal is co-localised with the CP within the lateral ventricles (Fig. 1c), providing evidence that it reflects BCSFB-mediated labelled blood water delivery to ventricular CSF. A good fit to the multi-TI BCSFB-ASL signal was observed using an adapted kinetic model for flow rate quantification (Supplementary Fig. 2), that returned a mean rate of $24 \pm 1$ ml/100 ml (of CSF)/min labelled water delivery to the lateral ventricles from the blood. In order to compare this non-invasive quantitative estimate of BCSFB function to historical measures of CP blood flow (which are expressed in units of ml/100 g (of CP tissue)/min), the brains of the same mice that were imaged underwent histological analysis to estimate the mass of the CP within the lateral ventricles (Supplementary Fig. 3). This returned a total CP mass of $0.22 \pm 0.01$ mg in the lateral ventricles, yielding a resultant CP blood flow of $1265 \pm 67$ ml/100 g/min. Using a standard multi-TI ASL model for CBF quantification in the cortex returned a perfusion rate of $283 \pm 12$ ml/100 g/min, in agreement with literature values[19,20]. Together, this is highly consistent with previous invasive measures in the rodent brain where CP blood flow is found to be ~5 times that of parenchymal blood flow[1,21,22].

A previous study, using invasive microsphere techniques, reported a marked decrease in CP blood flow following IV administration of anti-diuretic hormone vasopressin, with little change in parenchymal blood flow[23]. Reproducing these experiments, using the non-invasive techniques introduced here permitted baseline measures together with the response to vasopressin or vehicle in the same mice. Both traditional ASL

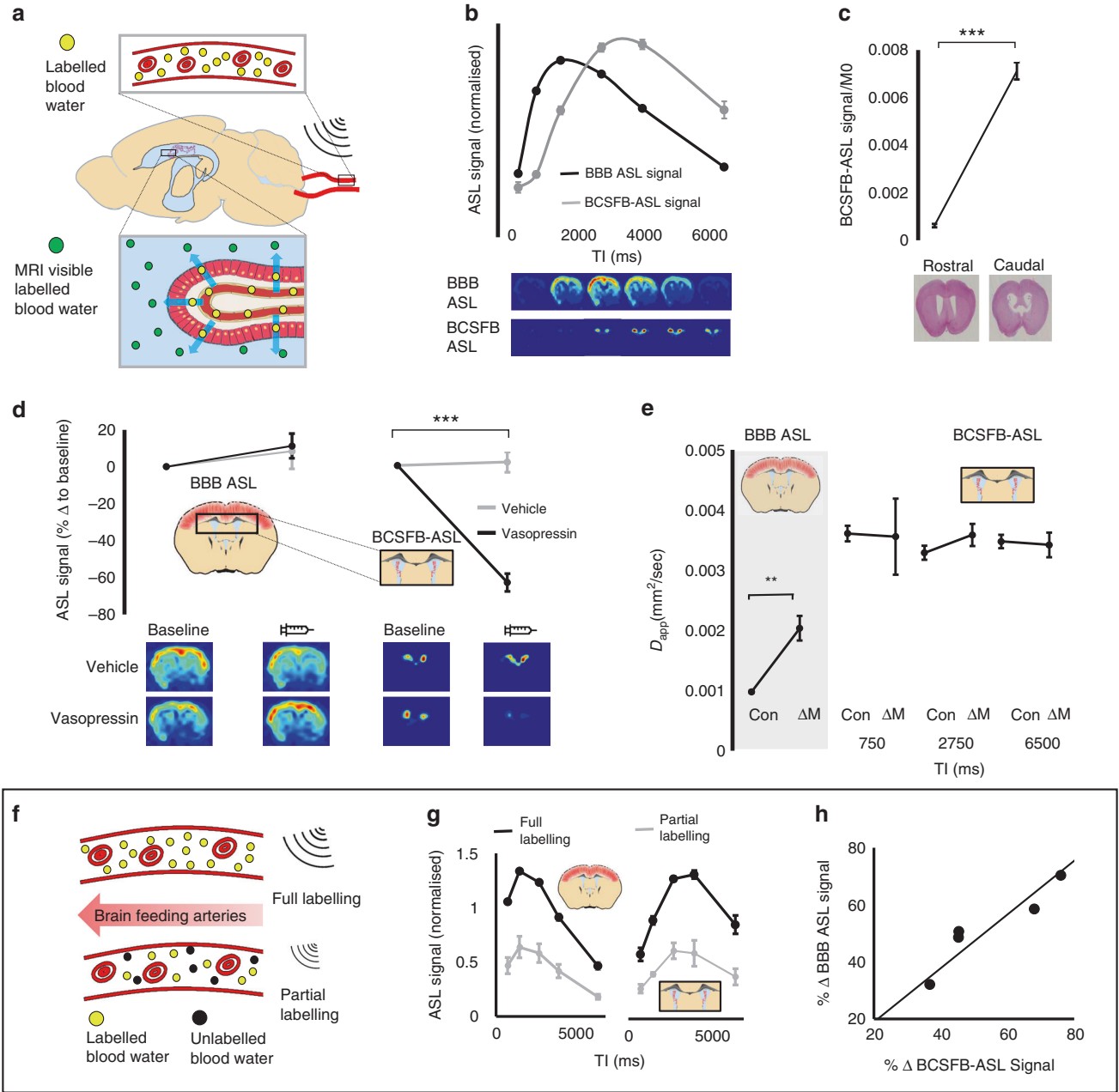

**Fig. 1 Non-invasive MRI of BCSFB function: characterisation and validation. a** Schematic: the technique works by labelling arterial blood water in the brain's feeding arteries and tuning the MRI acquisition (using TE = 220 ms) to measure only the signal from labelled blood water that has been transported to the CSF. **b** Above—the normalised traditional ASL signal that probes BBB function (taken from a cortical ROI) and novel BCSFB-ASL signal, as a function of TI ($n = 12$ biological independent animals examined over 12 independent experiments; error bars represent SEM); below—example ASL images from a single mouse at increasing TI for both techniques. **c** Normalised BCSFB-ASL signal (TI = 4 s) at the rostral and caudal section of the lateral ventricles. Example histological sections from rostral and caudal slices are shown for an example mouse that was imaged. $n = 10$; error bars represent SEM. ***$p = 0.000000014$ from a one-tailed $t$-test. **d** ASL signal (% relative to baseline) before and after administration (100 μU/ml, I.P 0.1 ml) of vasopressin or saline solution. Right column: standard ASL signal probing BBB function; left column: BCSFB-ASL signal. $n = 4/5$ biological independent animals examined over 4/5 independent experiments for vehicle and vasopressin, respectively; error bars represent SEM. **e** The pseudo-coefficient ($D_{app}$) of the standard ASL and control signal (grey box) and the BCSFB-ASL and control signal at three inflow times (TIs). $n = 4$ biological independent animals examined over four independent experiments; error bars represent SEM. **$p = 0.008$ from a one-tailed $t$-test. **f** Schematic illustrating modulation of arterial blood water labelling efficiency, the experimental strategy used for the data presented in **g** and **h**. **g** Measured normalised BBB (1st column) and BCSFB (2nd column) ASL signal as a function of TI with full (black line) and partial (grey line) labelling efficiency. $n = 5$ biological independent animals examined over five independent experiments; error bars represent SEM. **h** The % decrease in labelling efficiency for the BBB ($y$-axis) and BCSFB ($x$-axis) ASL signal across the five mice ($p = 0.02$, Pearson's correlation analysis, two sided).

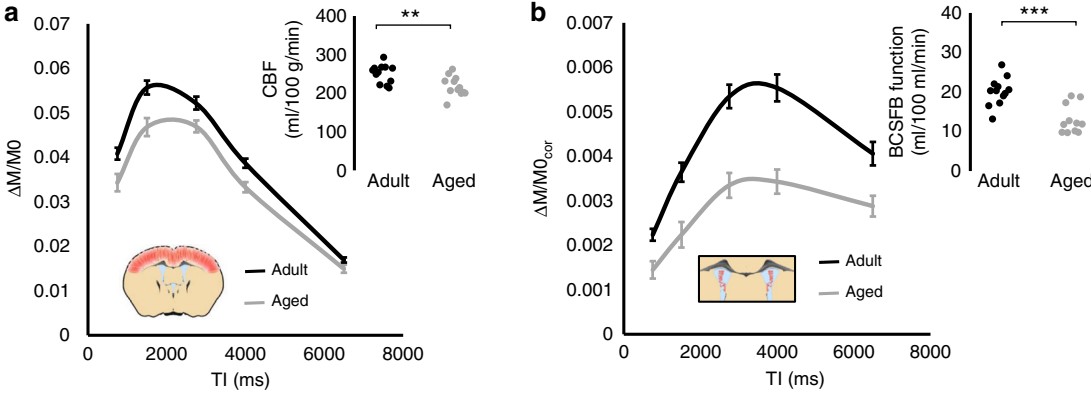

**Fig. 2 Non-invasive MRI of BCSFB function: application to the ageing brain. a** The normalised cortical ASL signal as a function of TI for the adult (black) and aged (grey) cohort. Error bars represent SEM across the group; inset: the estimated cortical blood flow for each individual subject. $n = 12/11$ biological independent animals examined over 12/11 independent experiments for each group, respectively. **$p = 0.0021$ from a one-tailed $t$-test. **b** The normalised lateral ventricle BCSFB-ASL signal as a function of TI for the aged (grey) and adult (black) cohort; error bars represent SEM across the group. Inset: the estimated rate of BCSFB-medicated blood water delivery to the lateral ventricles for each individual subject. $n = 12$ and 11 biological independent animals examined over 12 and 11 independent experiments for the adult and aged group, respectively. ***$p = 0.00006$ from a one-tailed $t$-test.

(that probes BBB function) and BCSFB-ASL measurements were acquired in the same animals (Fig. 1f). We recorded a 63% reduction in the BCSFB-ASL signal following vasopressin ($p < 0.001$) with no change recorded after vehicle. We recorded no change in relative parenchymal blood flow after vasopressin or vehicle using standard ASL, demonstrating the specificity of the method as a correlate of BCSFB function (Fig. 1d).

In order to further ensure that the BCSFB-ASL signal represents labelled blood water that has been delivered to ventricular CSF, with no contamination from labelled blood water in the vasculature, a hybrid diffusion-ASL MRI method was used to measure the pseudo-diffusion coefficient ($D_{app}$) of the ASL signals (Fig. 1e). As described in the intra-voxel-incoherent-motion MRI literature[24], $D_{app}$ is highly sensitive to the presence of intra-vascular water signal contributions. As hypothesised based on previous measures[25,26], the standard parenchymal ASL signal had markedly greater $D_{app}$ than the $D_{app}$ of parenchymal tissue (taken from the corresponding control signal) due to the contribution of labelled blood water in the blood vessels ($p < 0.001$). In contrast, and as hypothesised, the recorded $D_{app}$ of the BCSFB-ASL signal was highly similar to that of the CSF (taken from the control images at TE = 220 ms) at three separate TIs, providing evidence that the BCSFB-ASL signal derives from labelled blood water in ventricular CSF, with negligible vascular contamination, as hypothesised.

The movement of CSF within the ventricles presents a potential confounder when assessing BCSFB function using the ASL based technique proposed here. In order to rule out this possible confounder, the efficiency of arterial blood water labelling was modulated (Fig. 1f) using methods previously described in detail[27]. This resulted in a highly similar labelling-efficiency-driven reduction between the standard ASL signal (probing BBB function) and the BCSFB-ASL signal (Fig. 1g, h), demonstrating that both functional signals derive from a shared source: blood water, labelled in the brains feeding arteries, that has flowed into the brain cortex (BBB-ASL signal) or ventricular CSF (BCSFB-ASL signal), respectively, and does not reflect local CSF movement.

Together, these data provide comprehensive evidence that the method introduced here captures rates of BCSFB-mediated blood water delivery to ventricular CSF, a non-invasive surrogate measure of BCSFB function.

**Non-Invasive MRI of BCSFB Function: Application to the Ageing Brain.** Previous histological and in vitro data have recorded evidence of BCSFB dysfunction in the ageing brain[14,15]. Given the urgent need for safe and non-invasive early imaging biomarkers of age-related neurodegenerative processes, we applied the novel BCSFB-ASL method to a cohort of aged (23 months, $n = 12$, 33 ± 3 g) and adult mice (6 months, $n = 12$, 29 ± 2 g)—Fig. 2. Measures of cortical perfusion were acquired using standard multi-TI ASL techniques in the same imaging session. A 13% reduction in cortical perfusion was observed in the aged cohort (251 ± 7 ml/100 g/min (adult) vs. 218 ± 7 ml/100 g/min (aged), $p = 0.004$), in agreement with previous reports of cortical hypoperfusion in the ageing brain (Fig. 2a). A 36% reduction in rates of BCSFB-mediated blood water delivery to the ventricles was recorded (~20 ± 1 ml/100 ml/min vs. 13 ± 1 ml/100 ml/min, $p = 0.00004$), suggesting that BCSFB function may be especially vulnerable to the ageing process (Fig. 2b). The mass of CP tissue in the lateral ventricle of 24–25-month-old mice was 0.20 g ± 0.3 g yielding an estimated CP perfusion of 740 ± 60 ml/100 g/min. The aged CP tissue exhibited clear morphological changes (Supplementary Fig. 4) consistent with previous microscopy characterisation in the rodent brain[15].

No age-dependant changes in the volume of the lateral ventricles, the % volume of the lateral ventricles relative to whole brain volume or the T1 of the CSF were detected, with a subtle decrease in cortical T1 observed in the aged mouse brain ($p < 0.01$, Supplementary Fig. 5).

**Measuring the Action of the BCSFB to Shape Ventricular Volume.** CSF secretion from the BCSFB creates hydrostatic pressure gradients that play a role in determining ventricular volume[1]. Based on this, together with evidence from the literature that rates of CP water secretion are coupled to CP perfusion[28], we hypothesised a positive correlation between the volume of the lateral ventricles and the measure of BCSFB function in the adult mouse brain. A positive correlation between lateral ventricle volume and rates of BCSFB-mediated blood water delivery to the CSF was recorded across the two different sub-strains of C56BL/6 adult mice imaged in this study (Fig. 3a, b, C57BL/6J, female, ($n = 12$) $p = 0.02$; C57BL/6JRj, male, ($n = 12$) $p = 0.0001$). No correlation was observed in the aged C57BL/6JRj cohort, likely

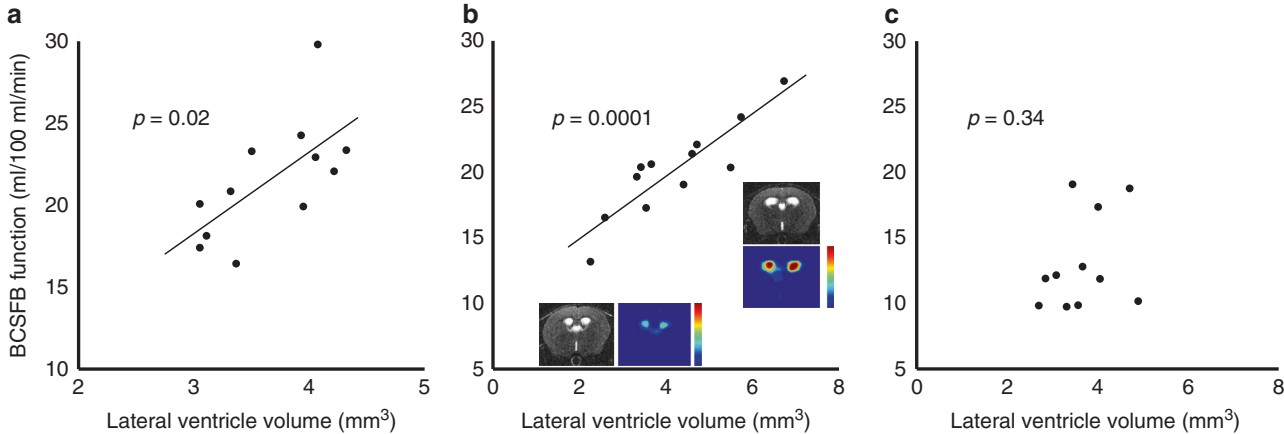

**Fig. 3 Non-invasive measures of BCSFB function correlate to ventricle volume in the adult mouse brain.** Scatter plots showing the estimated rate of BCSFB-mediated blood water delivery to the lateral ventricles against lateral ventricle volume for the multi-TI C57BL/6J (n = 12] 3-month adult female cohort (**a**), the 6-month adult C57BL/6JRj (n = 12) cohort (**b**) and the aged, 23-month C57Bl/6JRj (n = 11) cohort (**c**). Inset: example structural and functional BCSFB-ASL images for two C57/BL6J adult mice. P values from Pearson's correlation analysis, two sided.

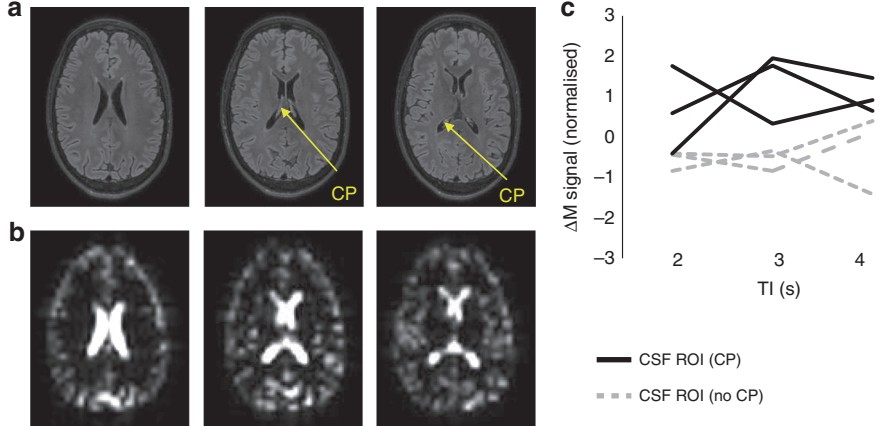

**Fig. 4 Non-invasive MRI of BCSFB function: proof of application to the human brain at 3T. a** High resolution FLAIR structural images showing the location of CP tissue within the lateral ventricles. **b** Corresponding axial control ASL images at TE = 400 ms. **c** The mean ASL signal (normalised to the average ASL signal in CP containing ROIs) as a function of TI for CSF regions with (black line) and without (grey line) CP tissue. Each line represents an individual subject (n = 3 biological replicates on three healthy volunteers over three separate experiments).

owing to the concomitant effects of unknown changes to CSF absorption pathways with ageing (Fig. 3c). This finding was reproduced in a separate cohort of aged mice (C57BL/6J, male, age: 24–25 months, n = 10)—Supplementary Fig. 6.

In support of this finding, a scan–rescan reproducibility study was performed in a separate cohort of six adult mice (Supplementary Fig. 7), that, for individual mice, returned a mean difference of 1.3 ml/100 ml/min in rates of BCSFB-mediated water delivery to the lateral ventricles (24 h between measurements [mean baseline = 21.1 ± 2.4 ml/100 ml/min; +24 h = 21.7 ± 2.6 ml/100 ml/min]). This high scan–rescan reproducibility demonstrates that the observed range in BCSBF function in Fig. 3a, b is primarily physiological, rather than methodological, in origin. Together, these data provide evidence that we are now able to capture a meaningful correlate of the mechanistic action of the BCSFB to shape ventricular volume, using non-invasive MRI.

**Non-Invasive MRI of BCSFB Function: Application to the Human Brain at 3T.** A proof-of-application MRI study was performed in three healthy volunteers to investigate method

feasibility in the human brain at 3T. 3D FLAIR structural images clearly identify the main locus of CP tissue within the lateral ventricles (Fig. 4a). Employing an echo time of 400 ms demonstrated suppression of tissue voxels in the control ASL images, with bright MRI signal now arising from CSF compartments (Fig. 4b). Across the three subjects, the ASL signals within regions of the CSF that contain CP tissue were greater than CSF regions without CP tissue (Fig. 4c). This demonstrates proof of application, albeit in a small cohort, together with preliminary evidence that the technique may have successfully captured a component of blood water delivery to the lateral ventricles associated with the BCSFB, in the human brain.

## Discussion

The BCSFB performs multiple actions critical to brain vitality and function: it regulates acid-base balance, removes waste products of neuronal/glial metabolism and transports nutrients, hormones, neurotransmitters and other neuropeptides to their eventual destination in the CNS[29]. Entwined to this functionality is the role of the BCSFB as the brain's primary source of CSF replenishment. Given these unique capabilities, BCSFB dysfunction

likely contributes to the development of many common brain disorders. Yet, research targeting the role of BCSFB function has been relatively scarce, owing to the lack of a practical and quantitative measurement technique analogous to traditional ASL or BOLD fMRI assessment of BBB function. Indeed, invasive assessment of BCSFB function in animal models using microscopy techniques is hampered by the deep brain location of CP tissue (relative to the assessment of blood vessels in the cerebral cortex for example)[21]. In this study, we introduce a non-invasive method for the assessment of BSCFB function, by quantifying rates of BCSFB-mediated water delivery from arterial blood to ventricular CSF using non-contrast MRI.

Previous studies, employing invasive surgical techniques, have measured reduced CSF secretion or CSF pressure in human ageing and Alzheimer's disease[30–33]. These observations are concordant with histological and in vitro studies of aged CP tissue from rat and sheep brains, respectively[14,15]. Applying the non-invasive technique introduced here to a mouse model of ageing (under isoflurane anaesthesia), we record a 36% reduction in rates of BCSFB-mediated water delivery relative to a more subtle decrease in parenchymal blood flow of 13% (Fig. 2). The mice imaged were 6 and 23 months old which is equivalent to ~30 and 65 human years[34]. Given that ageing is the primary risk factor for dementia and that cerebral perfusion is a sensitive biomarker of upstream neurodegenerative processes[35], this finding suggests that pathophysiologically relevant changes to BCSFB function may play an important role in the initiating stages of neurodegenerative disease. Indeed, CP dysfunction and concomitant reduction in CSF turnover impairs clearance of toxic metabolites and promotes neuro-inflammation, key pathophysiological correlates of early AD progression[36]. This non-invasive and translational method opens the door to practical study of BCSFB function in humans and repeated measures in animal models in order to more fully understand the role of BCSFB dysfunction in age-related cognitive decline. A previous study that used a gadolinium contrast agent to assess CP function in the human brain provided evidence for a reduction in CP permeability and perfusion with ageing[37]. Whilst, the application of this dynamic susceptibility contrast (DSC) technique is likely to be confounded by partial volume effects in the mouse brain (due to the size of CP tissue relative to voxel sizes typically used for DSC imaging), it would be valuable to compare this approach to the BCSFB-ASL technique in future applications to the human brain.

CSF secretion from the BCSFB creates hydrostatic pressure gradients thought to play an important role in the development and maintenance of the CNS[1]. Too little may hinder development and too much may result in hydrocephalus[38,39]. This is exemplified by the classical experiments of Dandy where unilateral choroid plexectomy resulted in collapse of the corresponding lateral ventricle[40]. On an individual patient level, however, the precise mechanisms that underlie ventriculomegaly remain elusive. Just as in the healthy adult human brain[41] the adult mice studied in this work present with a range of lateral ventricle sizes (as previously observed in the mouse brain[42,43]). This provided an opportunity to test a hypothesis that our measures of BCSFB function would correlate to ventricular volume, demonstrating that we are able to capture the action of the BSCFB to shape ventricular morphology. We observed strong correlation of BCSFB function to ventricle size in two separate cohorts of adult WT mice of different sub-strains (Fig. 3), which, in turn, provides evidence that our measures of BCSFB function represent a meaningful correlate of CSF secretion (likely reflecting known coupling of rates of CSF secretion to CP perfusion[28]). These findings are supported by good reproducibility of the measurements following scan–rescan analysis (Supplementary Fig. 7). Conversely, we did not observe a correlation between BCSFB

function and ventricular volume in the aged cohort (Fig. 3, Supplementary Fig. 6). This suggests that correlations between BCSFB function and ventricular volume can be captured in the young adult brain but this relationship may dissociate with age due the concomitant effects of age-related pathology of the CP (Fig. 2) in addition to possible age-related changes in CSF absorption pathways (which remain unknown due to the current lack of quantitative measurement techniques). In this work, a relatively large slice thickness was employed in the structural scans that were used to estimate ventricular volume. Despite this, reasonable precision was recorded from the scan–rescan reproducibility study (mean absolute error $0.2\ mm^3$ in $3\ mm^3$ mean volume). However, future studies may wish to employ a thinner slice thickness, or 3D acquisition, for more precise estimation of ventricular volume in the mouse brain. Interestingly, a previous clinical study reported increased CP volume in idiopathic intracranial hypertension patients and that decreased CP volume was associated with decreased ventricular volume following lumbar puncture[44]. The method introduced in this work can be applied to better understand the role of the BCSFB in CNS development in addition to brain disorders defined by ventriculomegaly such as hydrocephalus, together with functional modulation by novel therapy. Indeed, it would be interesting to apply the technique to mouse models of hydrocephalus to investigate possible changes to normal BCSFB function in these conditions.

The use of standard ASL methods to measure choroid plexus blood flow is likely to be readily confounded by partial volume effects owing to its proximity to highly perfused parenchymal tissue and the minimally 'perfused' CSF together with its relatively unpredictable location within the ventricular system. With standard ASL, it may be possible to position a ROI that contains largely CP tissue, but this will only reflect a fraction of choroid plexus blood flow within that ventricle. To overcome this, here, we take a new approach and use an ultra-long TE to ensure that only signal from the CSF is measured in the control/labelled images, with the ASL signal now deriving from labelled blood water that has crossed the BCSFB into the CSF (the BCSFB-ASL signal). By taking a large ROI that spans the lateral ventricles, we then estimate the average rate of labelled water delivery to the CSF across the entire lateral ventricles. In the mouse brain, the method requires two very simple adjustments to standard ASL sequences used to image parenchymal perfusion (increasing TE and voxel size) and thus, in principle, this approach can be easily implemented on clinical MRI scanners (the majority of which already have standard ASL sequences for CBF measurement), as demonstrated here (Fig. 4). The ultra-long TE isolates the signal from labelled blood water that has exchanged into the CSF, overcoming partial volume effects (that could confound the accuracy of blood water delivery rate quantification due to, e.g., proximity to the highly perfused parenchymal tissue) as only signal from the CSF is measured during the image readout. This means low resolution images can be acquired which markedly boosts sensitivity to the relatively small ASL signals that are detected, but without compromising the accuracy of quantification (Fig. 1). So here, for example, the mean BCSFB-ASL signal across the lateral ventricles that we measure is ~10% of the standard ASL signal that probes parenchymal perfusion (Fig. 2). The voxel size, however, is increased by a factor of ~12 from that used in typical mouse brain ASL imaging, so there is ~12 times more MRI signal per voxel. Therefore, the small signals that are detected are balanced by the increased voxel size, ensuring the method is technically feasible (see, e.g., Supplementary Fig. 2). Importantly, the low resolution imaging comes at little cost since, unlike imaging of the BBB where parenchymal vascular delivery often has high spatial affinity to the location of tissue metabolism, our measures of BCSFB function have little need for high spatial

resolution because the material delivered from the blood to the CSF is immediately dispersed around the ventricular compartment due to CSF pulsation. Moreover, this approach is likely to decrease measurement sensitivity to possible movement of the flexible and floating CP during the acquisition. Thus we anticipate that a global measure of BCSFB function in each ventricle, as reported for the lateral ventricles in this work, will represent a meaningful and practical measurement using this technique. In this work, our imaging slice was localised to the lateral ventricles given the large amount of CP tissue within this region. However, application to other CSF spaces that contain CP tissue (such as the third and fourth ventricle) should be straightforward. Moreover, a multi-slice or 3D version of the technique to simultaneously capture BCSFB function in CSF compartments across the whole brain should be readily achievable. Furthermore, the SNR efficiency of the technique could be further improved through the use of CASL, pCASL, time-encoded or multi-boli labelling schemes[45–47].

Given the small BCSFB-ASL signals that were detected in this study, it was important to carefully consider and rule out possible sources of artefact. Consequently, a series of experiments were carried out to determine whether the putative BCSFB-ASL signal derives from labelled blood water that has been delivered to the ventricles, as hypothesised, and thus represents a useful correlate of BCSFB function (Fig. 1). First, the control images demonstrate that only signal from the CSF is detected during image acquisition (indeed, given the T2 of tissue to be 38 ms[48], only 0.3% of the theoretical parenchymal tissue signal at TE = 0 ms remains at this echo time—Supplementary Fig. 1). Moreover, T2 of the traditional ASL signal at 9.4T in the mouse brain ranges in value from 20–33 ms at increasing TI and thus will be attenuated to an even greater degree[19]. Second, a wide slice-selective labelling width was implemented to ensure similar effect of the slice-selective and global flow-sensitive alternating inversion recovery (FAIR) pulse on the CSF compartments in the brain. Next, the dynamic time-series BCSFB-ASL data yielded a good fit (Supplementary Fig. 2) to an adapted 1-compartment (extra-vascular CSF) kinetic model that describes the delivery of blood water to the ventricular CSF (incorporating the T1 relaxation time of CSF measured from the multi-TI control data (Supplementary Fig. 1)). Furthermore, the ASL signal was markedly reduced at the rostral section of the lateral ventricles, consistent with the known location of CP tissue (confirmed by our own histological assessment). In order to rule out vascular contamination, the $D_{app}$ of the BCSFB-ASL signal was found to be highly similar to the $D_{app}$ of the CSF from the control data (unlike the traditional ASL signal where an increased $D_{app}$ was recorded, relative to the ADC of cortical brain tissue, as a portion of this signal is known to derive from labelled blood water in the vessels). We then modulated the efficiency of upstream arterial blood water labelling, observing that the effect on the measured BCSFB-ASL signal and the traditional ASL signal, respectively, was highly similar, demonstrating that these functional signals derive from a shared source: arterial blood water that has flowed into the brain (and not local CSF movement). Finally, we show that the technique can detect specific vasoconstriction of the CP following vasopressin, with no change recorded in parenchymal perfusion. Collectively, these data provide comprehensive validation for the accuracy of the technique to measure BCSFB-mediated blood water delivery to ventricular CSF, non-invasively.

In the context of this work, it is important to consider the difference between water exchange and net secretion[2,49]. Our quantitative estimates of the rate of labelled water delivery by the BCSFB to ventricular CSF do not represent a measure of CSF secretion, but primarily reflects the exchange of labelled blood water across the BCSFB. Thus, our measure reflects the rate of perfusion of the CP convolved with its permeability to labelled blood water. Given the huge CSF-facing surface area of the CP and its high vascular density, and based on the high extraction fraction of blood water in the brain tissue (~90%[22,50–52]), we surmise that 100% of labelled blood water crosses the CP to ventricular CSF. There is wide ranging evidence to suggest that the rate of CSF secretion is coupled to perfusion of the choroid plexus[28]. Thus, it is likely that the BCSFB-ASL measures in this work represent a correlate of CSF secretion. Indeed the observed correlations between our estimates of BCSFB function and ventricular volume suggest that this is the case (given wide ranging evidence that CSF secretion plays a role in determining ventricle size[53]). However, further studies that more fully explore this relationship using invasive measures of CSF secretion (e.g., Masserman or Papenheimer technique[2]) would be valuable. Taking the average rate of BCSFB-mediated labelled water delivery to the lateral ventricles and incorporating the total size of the functional voxels (11.25 mm³) returns a total BCSFB-labelled water delivery rate to the lateral ventricles of 2.7 μl/min. Unsurprisingly, this is markedly greater than previous estimates of CSF secretion in the mouse brain (~0.3 μl/min) which reflects higher rates of water exchange/flux vs. net secretion across blood vessels (see Hladky et al.[2] for detailed discussion). However, it is also important to consider that the previous estimates of CSF secretion will include contribution from CP tissue in the third and fourth ventricle as well as possible extra-choroidal sources such as the BBB[2] making it difficult to directly relate previous estimates of CSF secretion to rates of BCSFB-mediated labelled blood water delivery from the lateral ventricles that is estimated in this work.

Intuitively, the BCSFB measurement may be sensitive to transfer of labelled water back into CP tissue (having been delivered to the CSF). However, we believe this is unlikely to be a significant confounding factor to our current data interpretation, based on the following: (1) the volume of CP tissue relative to CSF in the lateral ventricles is ~5%, supporting the approximation of the CSF as a sink for labelled water; (2) the BCSFB-ASL signal shows a very good fit to an adapted 1-compartment Buxton kinetic model (Supplementary Fig. 2) that describes the unidirectional transfer of labelled water into the CSF compartment; (3) once labelled water is delivered to the CSF, the outer layers of the CP tissue will be less permeable to water relative to further diffusion/advection into free CSF.

In summary, here we introduce a method for non-invasive and quantitative assessment of BCSFB function, as an initiating step towards practical, repeated and comparable measures in the human brain and in animal models of pathology. Given the BCSFB's multifaceted functionality in support of brain homoeostasis, this method could have far-reaching implications in the study, diagnosis and treatment of brain disease.

## Methods

**BCSFB-ASL Theoretical Background.** The principle design of the technique introduced here is the use of an ASL MRI sequence but with an ultra-long TE (220 ms), low spatial resolution image readout. The ultra-long TE nulls the signal from the blood and parenchymal tissue whilst preserving signal from the CSF due to the high T2 relaxation time of CSF. Indeed, at this echo time, the signal from the grey matter tissue, blood and ventricular CSF will have decayed to ~0.3% and 0.07 and 33% of the theoretical signal at TE = 0, respectively (assuming T2s of 38, 30 and 200 ms for parenchyma, blood and CSF, respectively). The low spatial resolution readout enables the detection of the relatively small signals that derives from labelled blood water that has been delivered to the CSF compartment during the TI, due to the relatively large voxel size (analogous to the larger 'voxels' used in MRI spectroscopy to boost sensitivity). In this case, partial volume effects (which, in general, can represent an important possible confounder with low spatial resolution readouts in MRI) do not confound this approach because signal from the surrounding, highly perfused, parenchyma is eliminated by the ultra-long TE. Together, therefore, this sequence can reliably capture the signal from blood water that is labelled in the feeding arteries and crosses the BCSFB into the ventricular CSF

(giving the BCSFB-ASL signal). The data can then be analysed to yield a measure of mean BCSFB function across the lateral ventricles (see below).

**Mouse MRI—Animal Preparation and Anaesthesia.** A total of 72 mice were imaged in this study. All experiments were performed in accordance with the UK Home Office Animals (Scientific Procedures) Act. Anaesthesia was induced using 2% isoflurane in 0.4 L/min medical air and 0.1 L/min O$_2$ and was maintained at 2% isoflurane whilst the animal was placed on a MRI compatible plastic probe. The head was secured using ear bars, a bite bar and a nose cone to minimise motion during the data acquisition. Once the probe was fixed in the scanner, isoflurane concentration was reduced to 1.5% in 0.4 L/min medical air and 0.1 L/min O$_2$. Body core temperature was measured throughout using a rectal thermometer (Small Animal Instruments Inc) and maintained at 37 ± 0.5 °C using heated water tubing during the preparation and heated water tubing and warm air flow during the data acquisition period. Eye ointment was applied and breathing rate was monitored throughout the acquisitions using a respiration pillow sensor (Small Animal Instruments Inc). A scavenger pump was fixed inside the magnet bore to prevent build up of isoflurane.

All imaging was performed using a 9.4T VNMRS horizontal bore scanner (Agilent Inc, Palo Alto, CA). A 72 mm inner diameter volume coil was used for RF transmission and signal was received using a two channel array head coil (Rapid Biomedical).

**Multi-TI Traditional and BCSFB-ASL in the Lateral Ventricles (n = 12).** Twelve female C57/BL6J mice (3 months of age) were used in this study. Anatomical reference structural images were acquired with T2 weighting in order to clearly visualise the location of the major CSF compartments in the mouse brain (Fig. 1A) using a fast-spin echo, T2-weighted readout (FOV: 25 × 25 mm; matrix size = 256 × 256; echo train length = 8; TEeff = 48 ms; TR = 5 s, 14 slices, 0.5 mm slice thickness). For the ASL acquisitions, a 2.4 mm thick coronal slice was manually positioned so that it centred on the caudal aspect of the lateral ventricles (owing to the known location of the CP at the caudal aspect of the lateral ventricles[18]). Traditional ASL data were then acquired to estimate brain tissue perfusion using the following parameters: single slice; matrix size = 32 × 32; FOV = 20 × 20 mm; echo time = 20 ms. Separate acquisitions were performed at six TIs: 200, 750, 1500, 2750, 4000 and 6500 ms; number of repetitions = 5; repetition time = 12 s. A FAIR labelling scheme was employed with an adiabatic inversion pulse (bandwidth = 20 kHz) and a slice-selective labelling width of 19.2 mm. In order to capture rates of blood water delivery and exchange to ventricular CSF, via the CP, an identical ASL sequence was then applied but with two simple adjustments: the TE was extended to 220 ms and the number of repetitions increased to 20 (owing to the relatively low SNR of the ASL signal at long TE). A wide (19.2 mm) slice-selective labelling width was chosen to minimise possible ASL signal contamination due to differential effects of the global and slice-selective labelling pulse on the CSF compartments which in turn could lead to erroneous ASL signal due to CSF movement.

**Histology.** The brains of the mice imaged above (Multi-TI BCSFB-ASL in the Lateral Ventricles) were removed and fixed for histological analysis. The tissues were fixed in 10% formalin for 24 h and then processed and embedded into paraffin blocks. The mouse brains were orientated from rostral to caudal area, in order to obtain coronal sections. An initial assessment was performed to identify the location of the choroid plexus. Thereafter, 3 μm serial sections were cut and stained for Haematoxylin and Eosin. The brain slices sections were then scanned in an automated fashion using a NanoZoomer microscope and the area of the choroid plexus in the lateral ventricles was estimated by manual ROI drawing. The total volume was taken by calculating the area under the curve (Supplementary Fig. 3) and tissue density of 1 g/cm³ was assumed to convert the estimates of CP volume to CP mass.

**Traditional ASL Data Analysis.** For the traditional ASL data, given the large slice thickness and low spatial resolution, a single ROI was drawn in the cortex which is relatively homogenous within the imaging slice. For the data acquired at each of the TIs, ΔM images were generated by subtracting the labelled and control images in a pairwise fashion. M0 and the T1 of the cortical tissue were estimated by fitting the control signal to the standard inversion recovery model (Supplementary Fig. 1). The multi-TI ASL data were then fitted to the standard Buxton general kinetic model[54] to estimate cortical CBF.

**BCSFB-ASL Data Analysis.** The BCSFB-ASL images were obtained by subtracting the control and labelled images (TE = 220 ms) in a pairwise fashion. For each subject, ROIs were drawn that encompassed two regions of 3 × 2 voxels (12 voxels in total) that overlaid with the position of the lateral ventricles (see Supplementary Fig. 1). The sum of the signal was taken within this ROI for both the BCSFB-ASL images and control images. The control signal within the ROI was then used to calculate M0$^{CSF}$ and T1$^{CSF}$ by fitting the multi-TI data to a simple inversion recovery model (see Supplementary Fig. 1). In order to account for the volume of the lateral ventricles in the calculation of M0, the volume of the lateral ventricles was estimated by manual segmentation based on the high resolution anatomical reference images (acquired using the FSE structural sequence—see Supplementary

Fig. 1). Then a corrected M0 (M0$_{corr}$) was calculated by simply multiplying the M0 signal by the ratio of the size of the functional ROI (12 voxels—11.25 mm³) by the actual volume of the lateral ventricles (calculated from the high resolution anatomical reference scan (typically ~3–4 mm³)). This step is important for accurate quantification as, otherwise, the calculated M0 (a normalisation factor for CBF quantification) would be highly dependent on ventricle size due to partial volume effects in the low resolution ASL images. In this way, an estimate of the average rate of blood water delivery over the entirety of the lateral ventricles is calculated and reported in this work. The ΔM/M0$_{corr}$ signal as a function of TI was then fitted to a two compartment perfusion model first derived by Alsop and Detre[55] and later adapted by Wang et al.[56]. The model was then further adapted to describe transfer of labelled water from the blood and into the CSF, rather than into brain tissue (as in conventional ASL).

Under the condition that TI > δ:

$$\Delta M_{EV} = \frac{2M0 f \alpha}{\varphi} \left\{ \exp\left(-TI \, R_{1app}\right) [\exp(\min(TI, \, \delta + \tau)\Delta R) - \exp(\delta \Delta R)]/\Delta R] \right\},$$

$$(1)$$

where $f$ is the rate of delivery of labelled blood water to ventricular CSF (the quantitative surrogate marker of BCSFB function introduced in this work), $\tau$ is the temporal length of the labelled bolus of blood water, M0 is the equilibrium magnetisation (calculated from a fit of the control signal dependence on TI), $\varphi$ is the blood-CSF partition coefficient which was assumed to be 1, $R_{1app}$ is the apparent longitudinal relaxation of the CSF (approximated from a fit of the control signal dependence on TI), $R_{1a}$ is the longitudinal relaxation rate of blood (1/2.4 s based on a previous study[57] and $\Delta R = R_1 - R_{1a}$, $\delta$ is the tissue transit time (in this case the 'CSF transit time'). Here, extra-vascular $\Delta M_{EV}$ refers to measured ASL signal that derives from CSF. In this case the intra-vascular $\Delta M_{IV}$ component described in Wang et al.[56] is not included in the model owing to the ultra-long TE employed (220 ms—see above) that nulls the signal from blood. To quantify CP blood flow, we estimated the mass of the choroid plexus within the lateral ventricles (see above) and assumed an extraction fraction of 100% (given that BBB extraction fraction is ~90%[50] and the large surface area to volume ratio of the CP[6] combined with the high vascular density).

In order to compare the BCSFB functional signal recorded at the rostral section of the lateral ventricles, the slice position was positioned rostral by 2 mm and BCSFB-ASL data were captured at a single TI (4 s) in 10 of the 12 mice (in the 'Multi-TI Traditional and BCSFB-ASL in the Lateral Ventricles' experiments). Ventricular CSF was still present within the tagged and control images of these more rostral acquisitions. A paired t-test was used to compare the ΔM/M0$_{corr}$ values at the caudal and rostral sections of the lateral ventricles (acquired at TI = 4 s).

**Investigating the Effect of Reduced Labelling Efficiency on the ASL Signals (n = 5).** There is considerable movement of CSF in the brain (driven by a combination of CSF secretion by the CP, as well as cardiac and respiratory related pulsatility). This could represent a critical confounder when using ASL techniques if the slice-selective and global inversion pulse have a differential effect on CSF in the brain (resulting in inflow of labelled spins into the imaging slice during the inversion time, an effect analogous to the 'TimeSLIP' method to examine CSF movement[58]). In this study, we designed the MR sequence to minimise the effect of this possible confounder by using a wide slice selection inversion width (19.2 mm) designed to contain all the brain's CSF spaces. However, to provide further evidence that CSF movement is not confounding the data, here, we modulated the labelling efficiency and measured the effect on the standard ASL signal (that probes BBB function) and the novel BCSFB-ASL signal. Adult C57BL/6J mice (n = 5, female) were used for these experiments. Labelling efficiency was modulated by changing the bandwidth of the FAIR inversion pulse, as previously described in detail in our earlier work[27]. Images were acquired at five TIs (750, 1500, 2750, 4000 and 6500 ms) for standard ASL (5 repetitions) and BCSFB-ASL (20 repetitions) using a standard inversion pulse bandwidth of 20 kHz (full labelling) and then a bandwidth of 1 kHz (partial labelling). In one of the animals, data at two of the TIs were corrupted due to an unexpected hardware fault (only noise was present in the images) and therefore only data at three TIs were used to compare full and partial labelling. Given that we have shown the effect of decreasing the bandwidth of the inversion pulse on labelling efficiency to have a variable effect[27], we performed correlation analysis to compare the relative signal change between standard ASL and BCSFB-ASL from full to partial labelling across the five mice (Pearson's correlation coefficient). Here our hypothesis was that changing the labelling efficiency would have a highly similar effect on the standard ASL signal and the novel BCSFB-ASL signal, thus providing evidence that the measured BCSFB-ASL signal derives from labelled arterial blood water and not local CSF movement.

**Measuring the Pseudo-Diffusion Coefficient of the ASL Signals (n = 4).** A principle of the novel BSCFB-ASL technique is to measure the signal from blood water that has been labelled in the arteries and that has exchanged across the CP into the CSF space. Hence, unlike conventional ASL, the theoretical BCSFB-ASL signal does not include a contribution from the intra-vascular compartment (which will have almost completely decayed at TE = 220 ms @ 9.4T[19]). In these

experiments, a combined diffusion and ASL sequence was used to measure the pseudo-diffusion coefficient of the ASL signal ($D_{app}$). This is known to be sensitive to the proportion of intra- vs. extra-vascular labelled blood water[25,26]. This provides a means of examining whether the novel BCSFB-ASL signal, which we hypothesise to derive from labelled blood water that has been delivered to the lateral ventricles, is not contaminated by labelled blood water that is still in the vessels at the time of image capture. Thus, we hypothesised that the measured $D_{app}$ of the BCSFB-ASL signal would be no different to the $D_{app}$ of the control signal (which derives only from CSF owing to the TE employed (220 ms)). In contrast, for the standard ASL acquisitions, we hypothesised that the D* of the ASL signal would be greater than that of the control, owing to the contribution of labelled blood water in the vessels, as previously reported[26]. In order to test this hypothesis, experiments were performed on four female C57BL/6J mice (3 months of age). Standard ASL images (TE = 20 ms) were acquired at TI = 750 ms with and without motion probing gradients applied in the z-direction (b value = 200 s/mm²) using identical imaging parameters described above (Multi-TI Traditional and BCSFB-ASL in the Lateral Ventricles). BCSFB-ASL images (TE = 220 ms) were then acquired at TIs of 750, 2750 and 6500 ms with and without the same motion probing gradients (b value = 200 s/mm²) applied in the z-direction. For the traditional ASL data, both the ASL and control signal were taken within a cortical ROI and $D_{app}$ was calculated by fitting the ASL and control signal, respectively, as a function of b value to a simple mono-exponential model[25]. Similarly, the BCSFB-ASL signal was taken within a ROI across the lateral ventricles and $D_{app}$ was calculated in the same way for the ASL and control signal separately for the three separate TIs. In each case, a paired t-test was then used to investigate differences between the D* of the control and ASL signal.

**Vasopressin for Selective Modulation of Blood Flow to the Choroid Plexus (n = 10).** A previous study, using invasive microsphere techniques, reported that anti-diuretic hormone vasopressin markedly reduced blood flow to the CP, with little change to parenchymal perfusion[23]. This protocol provides a means to test the accuracy of the MRI technique introduced here to detect specific modulation of CP blood flow. For these experiments, a total of ten female C57BL/6J mice were used to reproduce this protocol using non-invasive MRI techniques. Here, standard ASL and BCSFB-ASL images were acquired at baseline (single TI (2/4 s), 5/20 averages, respectively), before and after administration of either vehicle (n = 4, saline 0.1 ml I.P) or vasopressin solution (n = 6, 100 µU/ml, 0.1 ml, I.P). Here, one outlier was identified and removed from subsequent analysis as the % change following vasopressin was outside 2 standard deviations of the mean (which we suspect was due to an error with the IP injection). ROIs in the cortex and lateral ventricles were drawn for the standard ASL and BCSFB-ASL data as described above and the % changes in ASL signal and BCSFB-ASL signal following vasopressin/vehicle from baseline was calculated.

**Application to the Ageing Mouse Brain (n = 24).** Aged C57BL/6JRj mice (23 months, n = 12, male) and sub-strain matched adult controls (6 months, n = 12, male) were supplied from Janvier labs (France). Mice underwent anaesthesia and MRI as described above (Multi-TI Measurements of CP Function in the Lateral Ventricles). A high resolution fast-spin echo T2-weighted structural scan was acquired for calculation of lateral ventricle volume using manual segmentation (fast-spin echo, T2-weighted readout (FOV: 25 × 25 mm; matrix size = 256 × 256; echo train length = 8; TEeff = 48 ms; TR = 5 s, 14 slices, 1 mm slice thickness). Standard ASL data (TE = 20, 5 averages) and BCSFB-ASL data (TE = 220 ms, 20 averages) were acquired at TIs of 750, 1500, 2750, 4000 and 6500 ms (TR = TI + 6 s). ASL data were quantified as described above and a t-test was used to investigate possible differences between 23 months and 6 months groups for the following parameters that were extracted from the imaging data: (1) the T1 of the cortical tissue and CSF, respectively (calculated by fitting the control signal (TE = 20 and 220 ms, respectively) as a function of TI to an inversion recovery model); (2) the volume of the lateral ventricles (from manual segmentation of the structural images (performed by an operator unaware of the corresponding functional data or animal group)); (3) the volume of the lateral ventricles as a % of whole brain volume (whole brain volume calculated by manual segmentation of the structural images); (4) cortical CBF; (5) the mean rate of blood water delivery to ventricular CSF (the quantitative surrogate marker of BCSFB function introduced in this work). In one of the aged mice, a MRI hardware failure lead to marked image corruption of functional data and therefore this data set was excluded from the data analysis. A separate cohort of aged mice (C57BL/6j mice, 24–25 months old, n = 6) was used for histological estimation of CP mass as described above.

**Investigating Correlations between BCSFB Function and Lateral Ventricle Volume.** The volume of the lateral ventricles was estimated by manual segmentation based on the high resolution, T2-weighted FSEMS images. Manuel segmentation was performed by an operator unaware of the corresponding functional data or animal group. Pearson's correlation analysis was then used to test for significant correlation between ventricular volume and rates of blood to CSF water delivery within the three groups of multi-TI BCSFB-ASL acquisitions performed in this study, respectively (C57/BL6J female 3 months (n = 12); C57BL/6JRj male

6 months (n = 12); C57BL/6JRj male 23 months (n = 11)). In order to further investigate age-related differences between correlations of BCSFB function and lateral ventricle volume, MRI studies were performed in a separate cohort of ten aged mice (C57BL/6j, male, 24–25 months old) using the parameters described above (Multi-TI BCSFB-ASL in the Lateral Ventricles) but with four TIs of 750, 1500, 4000 and 6500 ms.

**Scan–Rescan Reproducibility Study (n = 6).** Six female C57/BL6J mice (3 months of age) were used in this study. Measures of BCSFB function were captured using the identical parameters described above (Multi-TI BCSFB-ASL in the Lateral Ventricles) but with four TIs of 750, 1500, 4000 and 6500 ms. For each mouse, baseline scans were performed, the mouse was recovered and the same imaging protocol was applied 24 h later. The absolute difference in the estimated rate of BCSFB-mediated water delivery from the blood to the CSF in the lateral ventricles between the baseline and the 24 h follow-up scan was calculated for each mouse.

**Human Brain Imaging (n = 3).** Human brain MRI was performed on three healthy subjects (male, ages 27–34) using a 3T Philips Achieva scanner with a 32 channel head coil. A pseudo-CASL labelling duration of 3 s was applied and separate acquisitions were performed at post-labelling delays of 2, 3 and 4 s (TR = PLD + 6.5 s). Images were acquired across eight slices (slice thickness = 5 mm), with an in-plane matrix size of 64 × 64. A spin-echo EPI readout with a TE of 400 ms was used, with 25 averages acquired per TI. ROI analysis was performed and mean ASL signals were extracted within CSF regions deemed to contain CP tissue or be free of CP tissue, based on visual assessment of 3D T1-weighted FLAIR images acquired in the same subject (Fig. 4b). Human studies were approved by relevant ethical regulations and were performed on healthy volunteers with informed consent from all participants.

**Reporting Summary.** Further information on research design is available in the Nature Research Reporting Summary linked to this article.

## Data Availability

The raw imaging data are available for download via the UCL data repository library under the project 'Non-invasive MRI of blood–cerebrospinal fluid barrier function' (https://www.ucl.ac.uk/library/research-support/research-data-management/ucl-research-data-repository). https://doi.org/10.5522/04/12037521.v1. Source data underlying Figs. 1–4 are available as a Source Data file.

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

## Acknowledgements

J. A. W., P. G. E. and Y. O. are supported by the Wellcome Trust/Royal Society (204624/Z/16/Z). D. L. T. is supported by the UCL Leonard Wolfson Experimental Neurology Centre (PR/ylr/18575). This work is supported by the EPSRC-funded UCL Centre for Doctoral Training in Medical Imaging (EP/L016478/1) and the Department of Health's NIHR-funded Biomedical Research Centre at University College London Hospitals. M. F. L. receives funding from the EPSRC (EP/N034864/1); the King's College London and UCL Comprehensive Cancer Imaging Centre CR-UK and EPSRC, in association with the MRC and DoH (England); UK Regenerative Medicine Platform Safety Hub (MRC: MR/K026739/1). A. A. is supported by ECMC and CR-UK. The authors would like to thank Engineer Paul Hermans for his attention to detail and expertise in maintaining the Agilent 9.4T system. Finally, we would like to thank George Martin, Matthew Lawson, Lizzie Steptoe and Jayne Holby for their help in maintaining animal welfare and environmental enrichment.

## Author Contributions

P. G. E.: data acquisition, writing—review and editing. M. S.: data acquisition, methodology, writing—review and editing. A. A.: data acquisition, methodology, writing—review and editing. I. F. H.: technical expertise and training; writing—review and editing. Y. O.: writing—review and editing. P. N.–Y. O.: writing—review and editing. O. I.: writing—review and editing; E. M.: funding acquisition, supervision, writing—review and editing. M. F. L.: funding acquisition, supervision, methodology, writing—review and editing. D. L. T.: conceptualisation, data curation, methodology, writing—review and editing. J. A. W.: conceptualisation, resources, formal analysis, supervision, funding acquisition, investigation, visualisation, methodology, writing—original draft, project administration.

## Competing interests

The authors declare no competing interests.
