## [Peer Review File · Nature Communications]

Reviewers' comments:

Reviewer #1 (Remarks to the Author):

Precis:

In this elegant work by Evans and colleagues a novel non-invasive MRI method based on arterial spin labelling (ASL) MRI principles to measure brain-CSF-barrier (BCSFB) function is introduced. Arterial blood water is labeled but instead of using a short T2 (on 9.4T) which is typical of ASL for CBF measurements, an ultra-long TE (220ms) is implemented in the MRI sequence. This allows them to null out the signal from the blood and parenchymal tissue, to only capture transfer of labelled blood water in the choroidal capillaries into the CSF of the lateral ventricles at multiple inflow times TI. Using kinetic modeling, of the multi-TI BCSFB-ASL signal a measure of CSF or 'water delivery' across the vasculature of the choroid plexus (CP) is captured which is interpreted as a way to track or BCSFB function ("a non-invasive surrogate measure of BCSFB function'). The new BCSFB-ASL method is validated in various ways and the application to human brain is also investigated.

Major findings

- The BCSFB-ASL signal was co-localized to the CP of the lateral ventricles of the mouse brain and converted via kinetic modelling to $\sim 24\text{ml}/100\text{ml}$ CSF/min of water delivered to the lateral ventricles from CP blood. Converting this measure into CP blood flow (using total CP mass) CP blood flow of $\sim 1300\text{ml}/100\text{g}/\text{min}$ was derived – which is ~ 4 -fold higher than CBF of the mouse.
- Validation of technique for measuring BCSFB function:
 - o BCSFB from Caudal sections of the lateral ventricles \gg rostral sections i.e. signal is co-localized to the CP
 - o Vasopressin which is known to reduce CSF production lowers the BCSFB-ASL signal by 63%
 - o Intravascular water signal contributions to the BCSFB-ASL signal was excluded as a contaminant by measuring the pseudo diffusion coefficient (D^*) of the ASL signal.
 - o Local movement of CSF within ventricles is ruled out as a confounder by manipulating the arterial blood water labelling.
- Aging brain: While CBF is only reduced $\sim 10\%$, the BCSFB function is reduced by 21% in the aging mouse brain (23M) when compared to adult (6M)
- BCSFB function and ventricle volume is correlating in young adult (3M), adult (6M) but not in aging (23M) mice.
- Feasibility (proof-of-principle) of the BCSFB technique in live human brain (N=1) is provided showing increase of the ASL signal in CP rich areas (lateral ventricle) at 3T with TE=400ms over 3 different TIs; and relatively unchanged signal in a non-CP rich area.

Originality and significance: This work is truly original and represents a significant breakthrough in the ability to measure BCSFB function in the live rodent brain non-invasively. The technical aspects of the paper are well-explained and the execution and results provides assurance of rigor and reproducibility. The technique would have major impact in exploring the impact of a failing CP and (CSF production) on aspects of neurodegeneration and potentially also for understanding brain waste clearance. The potential for the technique to translate to the live human brain is also strong. Major strengths include 1) the quantitative conversion of the BCSFB function signal into CP flow in the mouse brain is in agreement with the literature previously reporting a much higher CP blood flow compared to CBF flow; 2) the effect of vasopressin on the BCSFB-function measure, 2) the ability to demonstrate a decrease of BCSFB function in aging mice; 4) strong evidence of reproducibility of the BCSFB functional measure over time.

Suggested improvements:

1. The relationship between BCSFB function and ventricle volume in the young adult and adult brain is convincing, but the lack of correlation between BCSFB function and ventricle volume in the aging brain is difficult to explain; because the ventricle volume is not any different than in the

young adult (Supplementary Fig. 5). One technical aspect to consider here is that volumetric measurements were (as far as this reviewer can tell) carried out using a 2D, anisotropic FSE T2-weighted sequence ($\sim 0.1 \text{ mm} \times 0.1 \text{ mm} \times 1 \text{ mm}$ = voxel size) which might have made the ventricle volume measurements inaccurate. It would be useful to have the ventricle volume measurements in relation to total brain volume to further assess the expected impact of aging on the brain (i.e. brain atrophy and ventricle enlargement).

2. The implication for the BCSFB function for hydrocephalus studies is very seductive given the data in Fig. 3, however, the concept fails with the lack of convincing data in aging brain. If the authors wish to stress the implication for hydrocephalus one possibility would be to apply the novel technique in a mouse model of hydrocephalus (e.g. transgenic or mutant mice models of ciliopathy).

3. It would be important to convert the BCSFB functional measure into CP flow in the aging population (i.e. similar to that done in the initial validation of the technique in the first cohort of mice).

4. could the authors comment on the direction of water transfer across the CP capillaries to CSF? i.e. how can they be sure that they are not measuring water transfer in the opposite direction also?

5. The human data could be strengthened by increasing the number of subjects slightly to see the ability to reproduce this signal change in CP rich and CP poor brain areas in more than one subject.

Conclusions: Truly novel and important work with high potential impact to the general field of CSF flow dynamics and neurodegeneration. The data/manuscript can be greatly strengthened by addressing suggested improvements noted above.

Reviewer #2 (Remarks to the Author):

Title: Non-invasive MRI of blood-cerebrospinal fluid barrier function

Overall: The structure and organization of this manuscript is excellent and it is technically sound. A very interesting technique to monitor BCSFB function non-invasively is presented and this technique will be useful for CSF-related disease mechanisms both in the rodents and in the human brain.

Comments and questions:

1) Choroid plexus (CP) capillary permeability and perfusion may be quantified by dynamic contrast-enhanced/dynamic susceptibility contrast enhanced MRI (DCE/DSC-MRI). Bouzerar et al., showed that CP capillary permeability and perfusion decreased with aging (Bouzerar R et al., *Neuroradiol* 2013, 55:1447–1454. doi: 10.1007/s00234-013-1290-2). Have the authors compared BCSFB-ASL and DCE/DSC-MRI? It would be nice to discuss the advantages of BCSFB-ASL compared to the ordinary DCE.

2) As the authors discussed in p11 line13, partial volume effects a concern since the choroid plexus (CP) is much smaller than the ventricles. The authors showed a clear correlation between BCSFB function and ventricular volume. Yet, CP size might also affect the result of BCSFB-ASL. Clinical literatures found a significant enlargement of ChPs compared with controls (Lublinsky S et al., *JMRI* 2018, 47:913–927. doi: 10.1002/jmri.25857) In addition, flexible and floating CP might move during the scan raising the issue of motion-induced artifacts. Have the authors experienced morphological changes during their experiments?

3) The correlation between BCSFB function and ventricle volume presented in Figure 3 is interesting. Have the authors considered intracranial pressure (ICP) or CSF pressure (CSFP) changes? Fleischman et al showed a significant reduction of CSF pressure with age and this literature might be strengthen the authors' hypothesis (Fleischman D et al., PLoS ONE 2012, 7:e52664. doi: 10.1371/journal.pone.0052664).

4) p20, line10,

In the methods part, the authors mentioned they used "high resolution T2 weighted FSEMS images" but do not include a description of the parameters and how much "high resolution" they used for the volume analysis. Is that the same protocol used in the anatomical reference images described in p15 line6-10? If yes, it is not "high resolution", since the slice thickness was 1mm. It would be a problematic to measure the ventricle volume using this resolution. Manual segmentation was used to calculate the ventricle volume. Additional details on how the segmentation was done and a discussion of the risk of bias should be included.

5) A quick estimate assuming 10 μ l of CSF in lateral ventricles results in \sim 2.4 μ l/min (p. 4, l. 11) of water extraction from CP in lateral ventricles, while invasive estimates of CSF secretion are about an order of magnitude lower (0.3-0.4 μ l/min) in mice. Can the authors describe in more detail the relationship between CP water exchange and CSF secretion and the mechanism that causes this difference?

6) The adapted kinetic model for quantifying the water delivery rate (p. 4, l. 11) does not appear to be described in the methods section.

7) To validate the specificity of the BCSFB-ASL method, it would be interesting to see BCSFB-ASL measurements from the entire brain — e.g. does water from large subarachnoid arteries exchange to the subarachnoid CSF space around it, e.g. around the circle of Willis? Does the sequence show signal in any locations other than large CSF volumes?

8) The authors interpret the very high correlation between ventricular size and BCSFB-water exchange as indicating that high water exchange is involved in enlarging ventricles (p. 10, l. 33). The authors argue that the long TE abolishes non-CSF signal, and thus partial volume effects should not affect the measurement. Conversely, the very high correlation could also be interpreted as a partial volume effect. Can the authors further substantiate their argument that no partial volume effect affects the measurements, e.g. by repeating experiments with incrementally decreasing voxel sizes to see if there's a correlation between voxel size and BCSFB-ASL signal?

9) BCSFB is mistyped several times as 'BSCFB'.

Reviewer #1

Originality and significance: This work is truly original and represents a significant breakthrough in the ability to measure BCSFB function in the live rodent brain non-invasively. The technical aspects of the paper are well-explained and the execution and results provides assurance of rigor and reproducibility. The technique would have major impact in exploring the impact of a failing CP and (CSF production) on aspects of neurodegeneration and potentially also for understanding brain waste clearance. The potential for the technique to translate to the live human brain is also strong. Major strengths include 1) the quantitative conversion of the BCSFB function signal into CP flow in the mouse brain is in agreement with the literature previously reporting a much higher CP blood flow compared to CBF flow; 2) the effect of vasopressin on the BCSFB-function measure, 2) the ability to demonstrate a decrease of BCSFB function in aging mice; 4) strong evidence of reproducibility of the BCSFB functional measure over time.

We thank the reviewer for their positive and constructive appraisal of the manuscript.

Suggested improvements:

1. The relationship between BCSFB function and ventricle volume in the young adult and adult brain is convincing, but the lack of correlation between BCSFB function and ventricle volume in the aging brain is difficult to explain; because the ventricle volume is not any different than in the young adult (Supplementary Fig. 5). One technical aspect to consider here is that volumetric measurements were (as far as this reviewer can tell) carried out using a 2D, anisotropic FSE T2-weighted sequence ($\sim 0.1 \text{ mm} \times 0.1 \text{ mm} \times 1 \text{ mm}$ = voxel size) which might have made the ventricle volume measurements inaccurate.

Following the reviewer's comment (and those of referee 2), we realised that some details regarding the parameters used for the volumetric measurements were unclear in the original submission and have sought to better clarify the parameters used to estimate ventricular volume in the revised manuscript. For the majority of the studies, a T2-weighted FSE sequence with a resolution of $\sim 0.1 \text{ mm} \times 0.1 \text{ mm} \times 0.5 \text{ mm}$ was employed. Example images from this sequence are shown below:

Using this sequence, the repeatability of ventricular volume measurements from the scan-rescan analysis was fairly good (mean absolute error 0.2 mm^3 [mean ventricular volume = 3.0 mm^3]). For the aged/adult mouse study presented in Figure 2, a thicker slice thickness was used (giving a resolution of $\sim 0.1 \text{ mm} \times 0.1 \text{ mm} \times 1 \text{ mm}$). Here, we agree with the reviewer that the structural imaging protocol (specifically the slice thickness) could have been better optimised for precise estimation of ventricular volume. However despite the limited precision of volume estimation, the acquisition and analysis were identical for the aged and adult data presented in Figure 2 and Figure 3B and 3C (adult and aged data captured in an interleaved manner). Thus, this technical aspect should not account for the relative lack of correlation between ventricular volume and the BCSFB-ASL signal in the aged mouse brain (relative to the younger brain). In addition, with regards to the data in Figure 3B, despite the non-optimal slice thickness (1mm), we are confident that the correlation is driven by genuine, marked, variation in ventricular volume across the adult cohort as illustrated by example data sets from 4 mice within this cohort, shown here:

Example Structural and Functional Images in the Adult Mouse Brain

Structural T2-w images (used to estimate the volume of the lateral ventricles [slice thickness =1 mm]) and accompanying BCSFB-ASL images at increasing TI (750-6500 ms) for four example C57BL/6j adult mice shown in Figure 2B. The BCSFB-ASL images use the same signal scale across the four mice shown here (illustrated by the colour bar on the RHS).

Despite the non-optimal slice thickness for precise ventricular volume estimation, we can reliably detect the marked variation in ventricular volume across the cohort that shows a striking correlation to independent functional measures of BCSFB function.

However, following this comment (and comment ii) below) and those of reviewer 2, we have acquired additional BCSFB-ASL data in a new cohort of aged mice (n=10, 23-24 months of age) with the higher resolution structural scan (see below). This important technical consideration is now also described in the discussion of the revised manuscript:

'In this work, a relatively large slice thickness was employed in the structural scans that were used to estimate ventricular volume. Despite this, reasonable precision was recorded from the scan-rescan reproducibility study (mean absolute error 0.2 mm³ in 3 mm³ mean volume). However, future studies may wish to employ a thinner slice thickness, or 3D acquisition, for more precise estimation of ventricular volume in the mouse brain.' Discussion (page 12).

It would be useful to have the ventricle volume measurements in relation to total brain volume to further assess the expected impact of aging on the brain (i.e. brain atrophy and ventricle enlargement).

Following the reviewer's suggestion, this has now been calculated and is reported in supplementary Figure 5 of the revised manuscript. No significant changes in % ventricular volume/whole brain volume was found, consistent with previous work that reported no loss of brain weight across the lifespan of a wild-type mouse, measured up to 23-26 months of age (Shimada 1999).

Supplementary Figure 5 d: Ventricle volume as a % of whole brain volume for the aged and adult cohort (d). Each dot represents an individual mouse.

2. The implication for the BCSFB function for hydrocephalus studies is very seductive given the data in Fig. 3, however, the concept fails with the lack of convincing data in aging brain.

Following the reviewer's comment, we have acquired additional data, designed to further investigate the findings presented in the original Figure 3. In addition, we have modified the text to clarify how the data presented in Figure 3 relates to possible future utility of the technique to hydrocephalus research.

BCSFB function was measured in an additional, separate, cohort of aged mice (male, C57BL/6j, 24-25 months, n=10). Here we employed a T2-w structural imaging protocol with a resolution of ~0.1 mm x 0.1 mm x 0.5 mm following comment i) above. In doing so, we recorded a mean estimate of BCSFB-mediated water delivery rates highly consistent with the original aged cohort (14.0 +/- 0.8 ml/100ml/min). Moreover, as hypothesized, we again recorded no significant correlation between ventricular volume and BCSFB function ($p = 0.19$).

Supplementary Figure 6: Rates of BCSFB-mediated blood water delivery to the ventricular CSF against the volume of the lateral ventricles for the separate cohort of aged mice (C57BL/6j, male, 24-25 months of age). Each dot represents each individual mouse (n=10).

As ventricle size reflects a balance between rates of CSF secretion and absorption, it is perhaps not surprising that in some cases [for example the aged mice examined here] there might not be a close association between BCSFB-ASL measurements and ventricle size, as unknown variations in CSF absorption pathways with ageing may be at play (unfortunately, however, there are no techniques able to quantify overall CSF absorption and so this remains unknown at this time). So while, at this early stage, we cannot fully explain the mechanisms underlying the relative lack of correlation in the aged brain, compared to the adult brain, that the reviewer highlights, this additional data provides reassurance that this a robust observation. These new data are shown below and are now incorporated into the revised manuscript (shown in supplementary Figure 6). Following the reviewer's comment we try to articulate this in the revised manuscript:

'No correlation was observed in the aged C57BL/6Jrj cohort, likely owing to the concomitant effects of age related impairment to BCSFB function and unknown changes to CSF absorption pathways with ageing (Figure 3c). This finding was reproduced in a separate cohort of aged mice (C57BL/6j, male, age: 24-25 months, n=10) – supplementary Figure 6'. Results (page 8).

'Conversely, we did not observe a correlation between BCSFB function and ventricular volume in the aged cohort (Figure 3). This suggests that correlations between BCSFB function and ventricular volume can be captured in the young adult brain but this relationship may dissociate with age due the concomitant effects of age-related pathology of the CP (Figure 2) in addition to possible age-related changes in CSF absorption pathways (which remain unknown due to the current lack of quantitative measurement techniques)'. Discussion (page 12).

If the authors wish to stress the implication for hydrocephalus one possibility would be to apply the novel technique in a mouse model of hydrocephalus (e.g. transgenic or mutant mice models of ciliopathy).

Our concern with using the new technique to investigate the relationship between BCSFB function and ventricular volume in a mouse model of hydrocephalus is that, to our knowledge, the existing mouse models of hydrocephalus are generated by inhibiting CSF clearance/absorption (e.g intra-ventricular kaolin injection or models of ciliopathy) rather than modulation of BCSFB function (not that these models necessarily reflect the aetiology of clinical hydrocephalus). Therefore, we did not have a well-founded hypothesis that our new measure of BCSFB function would be up or down-regulated in an existing mouse model of hydrocephalus (a hypothetical exception would be a mouse model of choroid plexus papilloma, but we were unable to find such a model in the literature). For this reason, we did not seek to apply the technique to a mouse model of hydrocephalus at the relative infancy of the development of the technique (given the unknown basis for a hypothesised change in BCSFB function). This contrasts to the ageing brain, for example, where several previous studies using invasive methods have reported downregulation of BCSFB function with ageing.

Given that ventricle size is a critical parameter in the diagnosis and management of hydrocephalus, in the original submission we aimed to communicate the implications of the findings presented in Figure 3 in the context of hydrocephalus to help stimulate ideas concerning future research/clinical application. However, based on the reviewers comment, we have sought to clarify this distinction (i.e we present evidence that we can capture a meaningful correlate of the action of the BCSFB to partially explain variations in ventricular size in the adult mouse brain but we are yet to demonstrate utility of the technique in conditions of hydrocephalus). We have removed any reference to hydrocephalus from the abstract and results that were present in the original manuscript.

‘Indeed, it would be interesting to apply the technique to mouse models of hydrocephalus to investigate possible changes to normal BCSFB function in these conditions.’ Discussion (page 12).

3. It would be important to convert the BCSFB functional measure into CP flow in the aging population (i.e. similar to that done in the initial validation of the technique in the first cohort of mice).

Following the reviewer’s comment, histological analysis was performed to estimate CP mass in the aged brain. Identical histological analysis was performed across a separate cohort of aged C57 BL/6j mice (24-25 months old, n = 6). A similar pattern of CP area from the rostral to caudal aspect of the lateral ventricles was recorded:

These new data are described in the revised manuscript:

‘The mass of CP tissue in the lateral ventricle of 24-25 month old mice was 0.20g ± 0.3 g yielding an estimated perfusion of 740 ± 60 ml/100g/min.’ Results (page 7).

4. could the authors comment on the direction of water transfer across the CP capillaries to CSF? i.e. how can they be sure that they are not measuring water transfer in the opposite direction also?

This intriguing aspect of water exchange and brain physiology underlying the measurement is now discussed in the revised manuscript:

‘Intuitively, the BCSFB measurement may be sensitive to transfer of labelled water back into CP tissue (having been delivered to the CSF). However, we believe this is unlikely to be a significant confounding factor to our current data interpretation, based on the following: i) The volume of CP tissue relative to CSF in the lateral ventricles is ~ 5%, supporting the approximation of the CSF as a sink for labelled water; ii) The BCSFB-ASL signal shows a very good fit to an adapted 1-compartment Buxton kinetic model (supplementary Figure 2) that describes the uni-directional transfer of labelled water into the CSF compartment; iii) Once labelled water is delivered to the CSF, the outer layers of the CP tissue will be less permeable to water relative to further diffusion/advection into free CSF.’ Discussion (page 15).

5. The human data could be strengthened by increasing the number of subjects slightly to see the ability to reproduce this signal change in CP rich and CP poor brain areas in more than one subject.

Following, the reviewer's suggestion, the protocol was applied to additional healthy volunteers. The new data reproduces the observed ASL signal increase in CP-rich vs CP-poor regions, albeit with variability in the shape of the dynamic time-courses at increasing TI, owing to the low SNR of the measurements at this early stage of development. Taking the average ASL signal (normalised to the mean ASL signal in the CP rich areas for each subject) across all the PLDs for each subject, there is a significant increase in CP rich vs CSF-only areas ($n = 3$, t-test: $p = 0.004$). The results are shown in a new version of Figure 4 in the revised manuscript:

Figure 4. Non-invasive MRI of BCSFB function: Proof of application to the human brain at 3T

- a) High resolution FLAIR structural images showing the location of CP tissue within the lateral ventricles
- b) Corresponding axial control ASL images at TE=400ms
- c) The mean ASL signal (normalised to the average ASL signal in CP containing ROIs) as a function of TI for CSF regions with (blackline) and without (grey line) CP tissue. Each line represents an individual subject ($n=3$).

Conclusions: Truly novel and important work with high potential impact to the general field of CSF flow dynamics and neurodegeneration. The data/manuscript can be greatly strengthened by addressing suggested improvements noted above.

Once again, we thank the reviewer for their constructive appraisal.

Reviewer #2

1) Choroid plexus (CP) capillary permeability and perfusion may be quantified by dynamic contrast-enhanced/dynamic susceptibility contrast enhanced MRI (DCE/DSC-MRI). Bouzerar et al., showed that CP capillary permeability and perfusion decreased with aging (Bouzerar R et al., *Neuroradiol* 2013, 55:1447–1454. doi: 10.1007/s00234-013-1290-2). Have the authors compared BCSFB-ASL and DCE/DSC-MRI? It would be nice to discuss the advantages of BCSFB-ASL compared to the ordinary DCE.

We thank the reviewer for raising this relevant study. Despite the higher spatial resolution that is typically employed with DCE/DSC techniques (relative to ASL), partial volume effects will be a significant confounding factor when applying these techniques to image CP function in the mouse brain. For example, taking the resolution (0.2mm x 0.2mm x 1mm employed by Pike *et al.*, for DCE/DSC in the mouse brain (Pike, Stoops et al. 2009)), and overlaying this on an example histology image:

One can see that it would be very challenging to separate the signal from the CP from the highly perfused parenchymal tissue (particularly with a slice thickness of 1mm), in addition to the difficulty of first-pass measurements in the mouse brain due to the very rapid circulation rates.

In regards to the use of such techniques in the human brain (as performed by Bouzerar and colleagues), we agree that this approach may provide a valuable comparison and this relevant study is now described in the revised manuscript:

'A previous study which used a gadolinium contrast agent to assess CP function in the human brain, provided evidence for a reduction in CP permeability and perfusion with ageing [Bouzerar et al.,]. Whilst the application of this dynamic contrast enhanced (DCE) technique is likely to be confounded by partial volume effects in the mouse brain (due to the size of CP tissue relative to voxel sizes typically used for DCE imaging), it would be valuable to compare this approach to the BCSFB-ASL technique in future application to the human brain'. Discussion (page 11).

2) As the authors discussed in p11 line13, partial volume effects a concern since the choroid plexus (CP) is much smaller than the ventricles. The authors showed a clear correlation between BCSFB function and ventricular volume. Yet, CP size might also affect the result of BCSFB-ASL. Clinical literatures found a significant enlargement of ChPs compared with controls (Lublinsky S et al., JMRI 2018, 47:913–927. doi: 10.1002/jmri.25857) In addition, flexible and floating CP might move during the scan raising the issue of motion-induced artifacts. Have the authors experienced morphological changes during their experiments?

We thank the reviewer for raising this important methodological consideration. Since the technique quantifies the overall rate of BCSFB-mediated delivery of labelled water to the lateral ventricles, it will reflect the mass of CP tissue, convolved with its perfusion [of blood water] and the extraction fraction to water. We believe that this is a useful characteristic of the method, since the measurement will then reflect the overall activity of the BCSFB within the lateral ventricular system (ie a global surrogate measure of the function of the BCSFB which we believe will be the most meaningful summary measure of its brain-wide functional role to, for example, stimulate brain clearance or drive substrate delivery though volume transmission).

Due to the difficulties of *in-vivo* structural imaging of the CP in the rodent brain (to our knowledge, there are no previous studies that report such measurements in the literature), we do not have a sense for possible morphological changes during the experiment. However, the possibility of the movement of the flexible and floating CP was a further motivation for the acquisition and analysis scheme used here in which we employ a very low spatial resolution single-shot readout (in-plane spatial resolution $\sim 0.8\text{mm}$, slice thickness = 2.4mm) and report the average BCSFB-mediated labelled water delivery across the entire lateral ventricles. In this sense, our approach is designed to minimise sensitivity to CP movement which is likely to significantly complicate functional assessment of the CP using traditional ASL methods, for example. This consideration is now described in the revised manuscript, along with the relevant findings of Lublinsky *et al.*:

'Importantly, the low resolution imaging comes at little cost since, unlike imaging of the BBB where parenchymal vascular delivery often has high spatial affinity to the location of tissue metabolism, our measures of BCSFB function have little need for high spatial resolution because the material delivered from the blood to the CSF is immediately dispersed around the ventricular compartment due to CSF pulsation. Moreover, this approach is likely to decrease measurement sensitivity to possible movement of the flexible and floating CP during the acquisition. Thus we anticipate that a global measure of BCSFB function in each ventricle, as reported for the lateral ventricles in this work, will represent a meaningful and practical measurement using this technique.' Discussion (page 13).

'Interestingly, a previous clinical study reported increased CP volume in idiopathic intracranial hypertension patients and that decreased CP volume was associated with decreased ventricular volume following lumbar puncture.' Discussion (page 12).

3) The correlation between BCSFB function and ventricle volume presented in Figure 3 is interesting. Have the authors considered intracranial pressure (ICP) or CSF pressure (CSFP) changes? Fleischman et al showed a significant reduction of CSF pressure with age and this literature might be strengthen the authors' hypothesis (Fleischman D et al., PLoS ONE 2012, 7:e52664. doi: 10.1371/journal.pone.0052664).

This relevant work is now discussed within the context of the data presented in Figure 2.

'Previous studies, employing invasive surgical techniques, have measured reduced CSF secretion or CSF pressure in human ageing and Alzheimer's disease (May, Kaye et al. 1990, Rubenstein 1998, Silverberg, Heit et al. 2001, Fleischman, Berdahl et al. 2012). These observations are concordant with histological and in-vitro studies of aged CP tissue from rat and sheep brains respectively (Serot, Foliguet et al. 2001, Chen, Kassem et al. 2009).' Discussion (page 11).

4) p20, line10,

In the methods part, the authors mentioned they used "high resolution T2 weighted FSEMS images" but do not include a description of the parameters and how much "high resolution" they used for the volume analysis. Is that the same protocol used in the anatomical reference images described in p15 line6-10? If yes, it is not "high resolution", since the slice thickness was 1mm. It would be a problematic to measure the ventricle volume using this resolution. Manual segmentation was used to calculate the ventricle volume. Additional details on how the segmentation was done and a discussion of the risk of bias should be included.

Following this comment, and those of reviewer 1, we have sought to better clarify the structural imaging parameters and analysis approach employed here (see above reply to reviewer 1). A discussion of the risk of bias is now included in the revised manuscript:

'Manual segmentation was performed by an operator unaware of the corresponding functional data or animal group (in the case of the aged vs adult comparison shown in Figure 2)'. Methods (page 21).

5) A quick estimate assuming 10 μ l of CSF in lateral ventricles results in \sim 2.4 μ l/min (p. 4, l. 11) of water extraction from CP in lateral ventricles, while invasive estimates of CSF secretion are about an order of magnitude lower (0.3-0.4 μ l/min) in mice. Can the authors describe in more detail the relationship between CP water exchange and CSF secretion and the mechanism that causes this difference?

As suggested by the reviewer, we now perform additional analysis to calculate the total rate of BCSFB water delivery to the lateral ventricles. This was done by simply multiplying the calculated mean rate of water delivery across the lateral ventricles (24 ml/100g/min) by the number of functional voxels over which the measurement was taken (12 voxels) and then multiplying by the total volume of the 12 functional voxels (in this way the measurement is independent of lateral ventricle size as intended). As estimated by the reviewer, this returns a rate of 2.7 μ l/min total BCSFB-mediated water delivery from the lateral ventricles. This important aspect of data interpretation is now described in the revised manuscript:

'Taking the average rate of BCSFB-mediated labelled water delivery to the lateral ventricles and incorporating the total size of the functional voxels (11.25mm³) returns a total BCSFB-labelled water delivery rate to the lateral ventricles of 2.7 μ l/min. Unsurprisingly, this is markedly greater than previous estimates of CSF secretion in the mouse brain (\sim 0.3 μ l/min) which reflects higher rates of water exchange/flux vs net secretion across blood vessels (see (Hladky and Barrand 2014) for detailed discussion). However, it is also important to consider that the previous estimates of CSF secretion will include contribution from CP tissue in the 3rd and 4th ventricle as well as possible extra-choroidal sources such as the BBB [2] making it difficult to directly relate previous estimates of CP secretion to rates of BCSFB-mediated labelled blood water delivery from the lateral ventricles that is estimated in this work'. Discussion (page 14).

6) The adapted kinetic model for quantifying the water delivery rate (p. 4, l. 11) does not appear to be described in the methods section.

The adapted model is described in the methods section under the subheading 'BCSFB-ASL Data Analysis' (Methods (page 18)).

7) To validate the specificity of the BCSFB-ASL method, it would be interesting to see BCSFB-ASL measurements from the entire brain — e.g. does water from large subarachnoid arteries exchange to the subarachnoid CSF space around it, e.g. around the circle of Willis? Does the sequence show signal in any locations other than large CSF volumes?

This is something that we were interested to see if we could observe with this acquisition. Throughout all the data capture, we have been carefully checking the BCSFB-ASL images to look for hotspots around other vessels proximal to the CSF. However, we could not find any reliable evidence for a reproducible ASL signal (@ TE = 220ms) in any other regions within the imaging slice other than the lateral ventricles (despite the fact that the coronal slice that was captured always contains subarachnoid arteries surrounded by CSF at the ventral aspect of the brain). A summary of

this analysis (from the multi-TI BCSFB-ASL data shown in Figure 1B, n = 12) is shown below where the mean BCSFB-ASL signal within a ROI around these subarachnoid arteries is 3% of the BCSFB-ASL signal in the lateral ventricles. Thus, this signal is too small to reliably capture, restricting interpretation of its possible physiological (or methodological) underpinnings at this moment in time.

This observation could be explained by the high flow rate and layers of smooth muscle/basement membrane within the large subarachnoid arteries. Conversely, the CP is made up of a highly vascularised capillary network with a huge CSF-facing surface area and as such, is exquisitely designed to interface the blood with the CSF, yielding relatively rapid labelled water exchange across the BCSFB.

8) The authors interpret the very high correlation between ventricular size and BCSFB-water exchange as indicating that high water exchange is involved in enlarging ventricles (p. 10, l. 33). The authors argue that the long TE abolishes non-CSF signal, and thus partial volume effects should not affect the measurement. Conversely, the very high correlation could also be interpreted as a partial volume effect. Can the authors further substantiate their argument that no partial volume effect affects the measurements, e.g. by repeating experiments with incrementally decreasing voxel sizes to see if there's a correlation between voxel size and BCSFB-ASL signal?

Following the reviewer's suggestion, the recommended experiments were performed. BCSFB-ASL data were acquired in a C57 BL/6 mouse (female, 4 months old) using an identical protocol as described in the manuscript (Multi-TI measurements) but with a single TI (4s) and 10 repetitions. The matrix size was then increased from 64 x 64 and then 96 x 96 and finally 128 x 128. This process was then repeated for a total of four measurements (10 repetitions each) at each matrix size in an interleaved manner. We explored two different analysis techniques: i) we manually segmented the lateral ventricles from the control images at each resolution; ii) we overlaid the same ROI described in the manuscript (a fixed volume ROI with 6 voxels around each lateral ventricle @32 x 32 matrix size; 24 voxels @ 64 x 64 matrix size; 48 voxels @ 96 x 96 matrix size; 96 voxels @ 128 x 128 matrix size). The data is summarised below:

Investigating the effect of different spatial resolutions on the BCSFB-ASL signal

- a.** Anatomical reference image showing the coronal slices contained within the functional imaging slice.
- b.** Control BCSFB-ASL images at increasing matrix size.
- c.** Corresponding BCSFB-ASL images at increasing matrix size.
- d.** Mean BCSFB/Control signal within ROI at increasing matrix size (each dot represents a single interleaved measurement (10 repetitions, ROI drawn round ventricles based on control images [illustrated schematically below for two example matrix sizes])). p-value is given for Pearson's correlation test.
- e.** Mean BCSFB/Control signal within ROI at increasing matrix size (each dot represents a single interleaved measurement (10 repetitions, fixed volume ROI used overlaid on BCSFB-ASL images [illustrated schematically below for two example matrix sizes])). p-value is given for Pearson's correlation test.

The $\Delta M/\text{control}$ signals display no evidence for dependence on the resolution of the functional images, providing reassurance that such partial volume effects are not affecting the measurements. This finding is consistent with the design of the acquisition where only CSF is captured with practically 0 signal recorded from blood and the surrounding tissue owing to the ultra-long TE employed (220ms).

9) BCSFB is mistyped several times as 'BSCFB'.

This error which has been corrected in the revised manuscript.

We thank the reviewer for their constructive appraisal of the manuscript.

Chen, R. L., N. A. Kassem, Z. B. Redzic, C. P. Chen, M. B. Segal and J. E. Preston (2009). "Age-related changes in choroid plexus and blood-cerebrospinal fluid barrier function in the sheep." Exp Gerontol **44**(4): 289-296.

Fleischman, D., J. P. Berdahl, J. Zaydlarova, S. Stinnett, M. P. Fautsch and R. R. Allingham (2012). "Cerebrospinal fluid pressure decreases with older age." PLoS One **7**(12): e52664.

Hladky, S. B. and M. A. Barrand (2014). "Mechanisms of fluid movement into, through and out of the brain: evaluation of the evidence." Fluids Barriers CNS **11**(1): 26.

May, C., J. A. Kaye, J. R. Atack, M. B. Schapiro, R. P. Friedland and S. I. Rapoport (1990). "Cerebrospinal fluid production is reduced in healthy aging." Neurology **40**(3 Pt 1): 500-503.

Pike, M. M., C. N. Stoops, C. P. Langford, N. S. Akella, L. B. Nabors and G. Y. Gillespie (2009). "High-resolution longitudinal assessment of flow and permeability in mouse glioma vasculature: Sequential small molecule and SPIO dynamic contrast agent MRI." Magn Reson Med **61**(3): 615-625.

Rubenstein, E. (1998). "Relationship of senescence of cerebrospinal fluid circulatory system to dementias of the aged." Lancet **351**(9098): 283-285.

Serot, J. M., B. Foliguet, M. C. Bene and G. C. Faure (2001). "Choroid plexus and ageing in rats: a morphometric and ultrastructural study." Eur J Neurosci **14**(5): 794-798.

Shimada, A. (1999). "Age-dependent cerebral atrophy and cognitive dysfunction in SAMP10 mice." Neurobiol Aging **20**(2): 125-136.

Silverberg, G. D., G. Heit, S. Huhn, R. A. Jaffe, S. D. Chang, H. Bronte-Stewart, E. Rubenstein, K. Possin and T. A. Saul (2001). "The cerebrospinal fluid production rate is reduced in dementia of the Alzheimer's type." Neurology **57**(10): 1763-1766.

REVIEWERS' COMMENTS:

Reviewer #1 (Remarks to the Author):

The revised work by Evans et al., has been greatly improved by the new supplementary data and edits in the new version of the manuscript. All of my concerns and comments have largely been addressed. In particular, new data again confirm the lack of correlation between BCFB-ASL measurement and ventricle size in the aging mouse brain. The human data have been supplemented and provide proof of principle. I have the following remaining minor comments that the authors should consider implementing:

- Pg 3 – the important and fundamental statement: “Ventricle size is determined by a fine balance between CSF secretion and absorption” must be followed by a reference with data showing that this is indeed the case. If there are no firm data supporting this statement it should be rephrased.
- Pg 7 – the statement: “CSF secretion from the BCSFB is known to create hydrostatic pressure gradients that play an important role in determining ventricular volume” should be followed by a reference. Again, it is a logical statement but might be more of a dogma with no data to support it in the mammalian brain. This sentence also appears on pg 11 in relation to development. Developmental data in relation to CSF and ventricles are interesting as it is known that the cerebral ventricles actually starts enlarging in the embryo before the CP is formed. There is a whole literature on the origin on the embryonic CSF prior to CP development. It might be worthwhile addressing or at least consider mentioning.
- Another possible artefact to consider is the anesthesia given that isoflurane was used (2%) and may affect CSF production (increases) which may change in the aging brain.

Reviewer #2 (Remarks to the Author):

The authors have responded powerfully to all my points critique. I recommend publication

Please find below a point-by-point response to the reviewer's final comments:

Reviewer #1 (Remarks to the Author):

The revised work by Evans et al., has been greatly improved by the new supplementary data and edits in the new version of the manuscript. All of my concerns and comments have largely been addressed. In particular, new data again confirm the lack of correlation between BCFB-ASL measurement and ventricle size in the aging mouse brain. The human data have been supplemented and provide proof of principle. I have the following remaining minor comments that the authors should consider implementing:

- Pg 3 – the important and fundamental statement: “Ventricle size is determined by a fine balance between CSF secretion and absorption” must be followed by a reference with data showing that this is indeed the case. If there are no firm data supporting this statement it should be rephrased.

We agree with the reviewer. We have edited the text and added a reference:

Page 3 (Introduction): ‘Ventricle size is thought to be determined by a balance between CSF secretion and absorption [Spector *et al.*, 2015].’

- Pg 7 – the statement: “CSF secretion from the BCSFB is known to create hydrostatic pressure gradients that play an important role in determining ventricular volume” should be followed by a reference. Again, it is a logical statement but might be more of a dogma with no data to support it in the mammalian brain. This sentence also appears on pg 11 in relation to development.

Again, we agree with the reviewer. We have edited the sentence and included a suitable reference.

Page 7 (Results): ‘CSF secretion from the BCSFB creates hydrostatic pressure gradients that play a role in determining ventricular volume [Spector *et al.*, 2015].’

Developmental data in relation to CSF and ventricles are interesting as it is known that the cerebral ventricles actually starts enlarging in the embryo before the CP is formed. There is a whole literature on the origin on the embryonic CSF prior to CP development. It might be worthwhile addressing or at least consider mentioning.

We have modified the text in light of the reviewer's comment:

Page 11(Discussion): ‘CSF secretion from the BCSFB creates hydrostatic pressure gradients thought to play an important role in the development and maintenance of the CNS [Spector *et al.*, 2015].’

- Another possible artefact to consider is the anesthesia given that isoflurane was used (2%) and may affect CSF production (increases) which may change in the aging brain.

We have modified the text in the discussion to place greater emphasis on the use of anaesthesia in these experiments:

Page 10 (Discussion): ‘Applying the non-invasive technique introduced here to a mouse model of ageing (under isoflurane anaesthesia), we record a 36% reduction in rates of BCSFB-mediated water delivery relative to a more subtle decrease in parenchymal blood flow of 13% (Figure 2).’

Reviewer #2 (Remarks to the Author):

The authors have responded powerfully to all my points critique. I recommend publication

Once again, we thank the reviewers for their constructive appraisal of the manuscript.